# Contrastive Learning as Goal-Conditioned Reinforcement Learning

Benjamin Eysenbach$^{\alpha,\beta}$    Tianjun Zhang$^{\gamma}$    Sergey Levine$^{\beta,\gamma}$    Ruslan Salakhutdinov$^{\alpha}$

$^{\alpha}$CMU        $^{\beta}$Google Research        $^{\gamma}$UC Berkeley

## Abstract

In reinforcement learning (RL), it is easier to solve a task if given a good representation. While *deep* RL should automatically acquire such good representations, prior work often finds that learning representations in an end-to-end fashion is unstable and instead equip RL algorithms with additional representation learning parts (e.g., auxiliary losses, data augmentation). How can we design RL algorithms that directly acquire good representations? In this paper, instead of adding representation learning parts to an existing RL algorithm, we show (contrastive) representation learning methods can be cast as RL algorithms in their own right. To do this, we build upon prior work and apply contrastive representation learning to action-labeled trajectories, in such a way that the (inner product of) learned representations exactly corresponds to a goal-conditioned value function. We use this idea to reinterpret a prior RL method as performing contrastive learning, and then use the idea to propose a much simpler method that achieves similar performance. Across a range of goal-conditioned RL tasks, we demonstrate that contrastive RL methods achieve higher success rates than prior non-contrastive methods, including in the offline RL setting. We also show that contrastive RL outperforms prior methods on image-based tasks, without using data augmentation or auxiliary objectives. [1]

## 1  Introduction

Representation learning is an integral part of reinforcement learning (RL[2]) algorithms. While such representations might emerge from end-to-end training [7, 79, 119, 126], prior work has found it necessary to equip RL algorithms with perception-specific loss functions [32, 44, 71, 89, 91, 101, 116, 140] or data augmentations [69, 73, 116, 118], effectively decoupling the representation learning problem from the reinforcement learning problem. Given what prior work has shown about RL in the presence of function approximation and state aliasing [2, 135, 138], it is not surprising that end-to-end learning of representations is fragile [69, 73]: an algorithm needs good representations to drive the learning of the RL algorithm, but the RL algorithm needs to drive the learning of good representations. So, *can we design RL algorithms that do learn good representations without the need for auxiliary perception losses?*

Rather than using a reinforcement learning algorithm also to solve a representation learning problem, we will use a representation learning algorithm to also solve certain types of reinforcement learning problems, namely goal-conditioned RL. Goal-conditioned RL is widely studied [6, 15, 22, 62, 80, 120], and intriguing from a representation learning perspective because it can be done in an entirely self-supervised manner, without manually-specified reward functions. We will focus on contrastive (representation) learning methods, using observations from the same trajectory (as done

---

[1]Project website with videos and code: `https://ben-eysenbach.github.io/contrastive_rl`

[2]RL = reinforcement learning, not representation learning.

36th Conference on Neural Information Processing Systems (NeurIPS 2022).

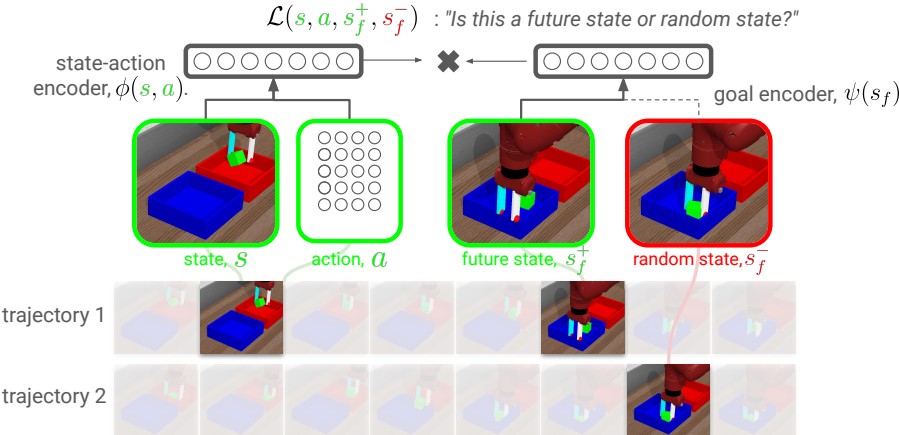

Figure 1: **Reinforcement learning via contrastive learning.** Our method uses contrastive learning to acquire representations of state-action pairs ($\phi(s, a)$) and future states ($\psi(s_f)$), so that the representations of future states are closer than the representations of random states. We prove that learned representation corresponds to a value function for a certain reward function. To select actions for reaching goal $s_g$, the policy chooses the action where $\phi(s, a)$ is closest to $\psi(s_g)$.

in prior work [95, 109]) while also including actions as an additional input (See Fig. 1). Intuitively, contrastive learning then resembles a goal-conditioned value function: nearby states have similar representations and unreachable states have different representations. We make this connection precise, showing that sampling positive pairs using the discounted state occupancy measure results in learning representations whose inner product exactly corresponds to a value function.

In this paper, we show how contrastive representation learning can be used to perform goal-conditioned RL. We formally relate the learned representations to reward maximization, showing that the inner product between representations corresponds to a value function. This framework of contrastive RL generalizes prior methods, such as C-learning [29], and suggests new goal-conditioned RL algorithms. One new method achieves performance similar to prior methods but is simpler; another method consistently outperforms the prior methods. On goal-conditioned RL tasks with image observations, contrastive RL methods outperform prior methods that employ data augmentation and auxiliary objectives, and do so without data augmentation or auxiliary objectives. In the offline setting, contrastive RL can outperform prior methods on benchmark goal-reaching tasks, sometimes by a wide margin.

## 2 Related Work

This paper will draw a connection between RL and contrastive representation learning, building upon a long line of contrastive learning methods in NLP and computer vision, and deep metric learning [17, 53, 54, 54, 56, 77, 84, 86, 87, 94, 95, 108, 109, 113, 122, 129, 132]. Contrastive learning methods learn representations such that similar ("positive") examples have similar representations and dissimilar ("negative") examples have dissimilar representations.[3] While most methods generate the "positive" examples via data augmentation, some methods generate similar examples using different camera viewpoints of the same scene [109, 122], or by sampling examples that occur close in time within time series data [4, 95, 109, 118]. Our analysis will focus on this latter strategy, as the dependence on time will allow us to draw a precise relationship with the time dependence in RL.

*Deep* RL algorithms promise to automatically learn good representations, in an end-to-end fashion. However, prior work has found it challenging to uphold this promise [7, 79, 119, 126], prompting many prior methods to employ separate objectives for representation learning and RL [32, 44, 71, 89, 91, 100, 101, 116, 118, 140, 143]. Many prior methods choose a representation learning objectives that reconstruct the input state [32, 47, 49, 50, 71, 91, 93, 141] while others use contrastive representation learning methods [89, 95, 111, 116, 118]. Unlike these prior methods, we will not use a separate representation learning objective, but instead use the same objective for both representation learning and reinforcement learning. Some prior RL methods have also used contrastive learning to acquire reward functions [14, 20, 33, 38, 63, 67, 92, 133, 134, 146], often in imitation learning

---

[3]Our focus will not be on recent methods that learn representations without negative samples [18, 43].

settings [37, 55]. In contrast, we will use contrastive learning to directly acquire a value function, which (unlike a reward function) can be used directly to take actions, without any additional RL.

This paper will focus on goal-conditioned RL problems, a problem prior work has approached using temporal difference learning [6, 29, 62, 80, 103, 106], conditional imitation learning [22, 41, 83, 105, 120], model-based methods [23, 107], hierarchical RL [90], and planning-based methods [30, 93, 105, 115]. The problems of automatically sampling goals and exploration [24, 35, 85, 98, 144] are orthogonal to this work. Like prior work, we will parametrize the value function as an inner product between learned representations [34, 58, 106]. Unlike these prior methods, we will learn a value function directly via contrastive learning, without using reward functions or TD learning.

Our analysis will be most similar to prior methods [11, 15, 29, 103] that view goal-conditioned RL as a data-driven problem, rather than as a reward-maximization problem. Many of these methods employ hindsight relabeling [6, 26, 62, 78], wherein experience is *relabeled* with an outcome that occurred in the future. Whereas hindsight relabeling is typically viewed as a trick to add on top of an RL algorithm, this paper can roughly be interpreted as showing that the hindsight relabeling is a standalone RL algorithm. Many goal-conditioned methods learn a value function that captures the similarity between two states [29, 62, 91, 125]. Such distance functions are structurally similar to the critic function learned for contrastive learning, a connection we make precisely in Sec. 4. In fact, our analysis shows that C-learning [29] is already performing contrastive learning, and our experiments show that alternative contrastive RL methods can be much simpler and achieve higher performance.

Prior work has studied how representations related to reward functions using the framework of universal value functions [12, 106] and successor features [9, 52, 81]. While these methods typically require additional supervision to drive representation learning (manually-specified reward functions or features), our method is more similar to prior work that estimates the discounted state occupancy measure as an inner product between learned representations [11, 131]. While these methods use temporal difference learning, ours is akin to Monte Carlo learning. While Monte Carlo learning is often (but not always [23]) perceived as less sampling efficient, our experiments find that our approach can be as sample efficient as TD methods. Other prior work has focused on learning representations that can be used for planning [59, 82, 104, 105, 128]. Our method will learn representations using an objective similar to prior work [105, 109], but makes the key observation that the representation already encodes a value function: no additional planning or RL is necessary to choose actions.

Please see Appendix A for a discussion of how our work relates to unsupervised skill learning.

## 3 Preliminaries

**Goal-conditioned reinforcement learning.** The goal-conditioned RL problem is defined by states $s_t \in \mathcal{S}$, actions $a_t$, an initial state distribution $p_0(s)$, the dynamics $p(s_{t+1} \mid s_t, a_t)$, a distribution over goals $p_g(s_g)$, and a reward function $r_g(s, a)$ for each goal. This problem is equivalent to a multi-task RL [5, 45, 121, 130, 139], where tasks correspond to reaching goals states. Following prior work [11, 15, 29, 103], we define the reward as the probability (density) of reaching the goal at the next time step:[4]

$$r_g(s_t, a_t) \triangleq (1 - \gamma)p(s_{t+1} = s_g \mid s_t, a_t). \tag{1}$$

This reward function is appealing because it avoids the need for a human user to specify a distance metric (unlike, e.g., [6]). Even though our method will not estimate the reward function, we will still use the reward function for analysis. For a goal-conditioned policy $\pi(a \mid s, s_g)$, we use $\pi(\tau \mid s_g)$ to denote the probability of sampling an infinite-length trajectory $\tau = (s_0, a_0, s_1, a_1, \cdots)$. We defined the expected reward objective and Q-function as

$$\max_\pi \mathbb{E}_{p_g(s_g),\pi(\tau|s_g)}\left[\sum_{t=0}^{\infty} \gamma^t r_g(s_t, a_t)\right], \quad Q^\pi_{s_g}(s, a) \triangleq \mathbb{E}_{\pi(\tau|s_g)}\left[\sum_{t'=t}^{\infty} \gamma^{t'-t} r_g(s_{t'}, a_{t'}) \mid {}^{s_t=s,}_{a_t=a}\right]. \tag{2}$$

Intuitively, this objective corresponds to sampling a goal $s_g$ and then optimizing the policy to go to that goal and stay there. Finally, we define the discounted state occupancy measure as [55, 142]

$$p^{\pi(\cdot|\cdot,s_g)}(s_{t+} = s) \triangleq (1 - \gamma)\sum_{t=0}^{\infty} \gamma^t p_t^{\pi(\cdot|\cdot,s_g)}(s_t = s), \tag{3}$$

---

[4]At the initial state, this reward also includes the probability that the agent started at the goal: $r_g(s_0, a_0) = (1 - \gamma)(p(s_1 = s_g \mid s_0, a_0) + p_0(s_0 = s_g))$

where $p_t^\pi(s)$ is the probability density over states that policy $\pi$ visits after $t$ steps. Sampling from the discounted state occupancy measure is easy: the first sample a time offset from a geometric distribution ($t \sim \text{GEOM}(1 - \gamma)$), and then look at what state the policy visits after exactly $t$ steps. We will use $s_{t+}$ to denote states sampled from the discounted state occupancy measure. Because our method will combine experience collected from multiple policies, we also define the average stationary distribution as $p^{\pi(\cdot|\cdot)}(s_{t+} = s \mid s, a) \triangleq \int p^{\pi(\cdot|\cdot,s_g)}(s_{t+} = s \mid s, a) p^\pi(s_g \mid s, a) ds_g$, where $p^\pi(s_g \mid s, a)$ is the probability of the *commanded* goal given the current state-action pair. This stationary distribution is equivalent to that of the policy $\pi(a \mid s) \triangleq \int \pi(a \mid s, s_g) p^\pi(s_g \mid s) ds_g$ [145].

**Contrastive representation learning.** Contrastive representation learning methods [17, 46, 53, 54, 61, 77, 84, 86, 87, 122, 124, 129] take as input pairs of positive and negative examples, and learn representations so that positive pairs have similar representations and negative pairs have dissimilar representations. We use $(u, v)$ to denote an input pair (e.g., $u$ is an image, and $v$ is an augmented version of that image). Positive examples are sampled from a joint distribution $p(u, v)$, while negative examples are sampled from the product of marginal distributions, $p(u)p(v)$. We will use an objective based on binary classification [77, 86, 87, 94]. Let $f(u, v) = \phi(u)^T \psi(v)$ be the similarity between the representations of $u$ and $v$. We will call $f$ the *critic function*[5] and note that its range is $(-\infty, \infty)$. We will use NCE-binary [84] objective (also known as InfoMAX [54]):

$$\max_{f(u,v)} \mathbb{E}_{\substack{(u,v^+) \sim p(u,v) \\ v^- \sim p(u)}} \left[ \log \sigma( \underbrace{f(u, v^+)}_{\phi(u)^T \psi(v^+)} ) + \log(1 - \sigma( \underbrace{f(u, v^-)}_{\phi(u)^T \psi(v^-)} )) \right]. \tag{4}$$

# 4 Contrastive Learning as an RL Algorithm

This section shows how to use contrastive representation to *directly* perform goal-conditioned RL. The key idea (Lemma 4.1) is that contrastive learning estimates the Q-function for a certain policy and reward function. To prove this result, we relate the Q-function to the state occupancy measure (Sec. 4.1) and then relate the optimal critic function to the state occupancy measure (Sec. 4.2).

This result allows us to propose a new algorithm for goal-conditioned RL based on contrastive learning. Unlike prior work, this algorithm is not adding contrastive learning on top of an existing RL algorithm. This framework generalizes C-learning [29], offering a cogent explanation for its good performance while also suggesting new methods that are simpler and can achieve higher performance.

## 4.1 Relating the Q-function to probabilities

This section sets the stage for the main results of this section by providing a probabilistic perspective goal-conditioned RL. The expected reward objective and associated Q-function in (Eq. 2) can equivalently be expressed as the probability (density) of reaching a goal in the future:

**Proposition 1** (rewards → probabilities)**.** *The Q-function for the goal-conditioned reward function* $r_g$ *(Eq. 1) is equivalent to the probability of state* $s_g$ *under the discounted state occupancy measure:*

$$Q_{s_g}^\pi(s, a) = p^{\pi(\cdot|\cdot,s_g)}(s_{t+} = s_g \mid s, a). \tag{5}$$

The proof is in Appendix B. Translating rewards into probabilities not only makes it easier to analyze the goal-conditioned problem, but also means that any method for estimating probabilities (e.g., contrastive learning) can be turned into a method for estimating this Q-function.

## 4.2 Contrastive Learning Estimates a Q-Function

We will use contrastive learning to learn a value function by carefully choosing the inputs $u$ and $v$. The first input, $u$, will correspond to a state-action pair, $u = (s_t, a_t) \sim p(s, a)$. In practice, these pairs are sampled from the replay buffer. Including the actions in the input is important because it will allow us to determine which actions to take to reach a desired future state. The second variable, $v$, is a *future* state, $v = s_f$. For the "positive" training pairs, the future state is sampled from the discounted

---

[5] In contrastive learning, the critic function indicates the similarity between a pair of inputs [99]; in RL, the critic function indicates the future expected returns [66]. Our method combines contrastive learning and RL in a way that these meanings become one and the same.

state occupancy measure, $s_f \sim p^{\pi(\cdot|\cdot)}(s_{t+} \mid s_t, a_t)$. For the "negative" training pairs, we sample a future state from a random state-action pair: $s_f \sim p(s_{t+}) \triangleq \int p^{\pi(\cdot|\cdot)}(s_{t+} \mid s, a)p(s, a)dsda$. With these inputs, the contrastive learning objective (Eq. 4) can be written as

$$\max_f \mathbb{E}_{\substack{(s,a)\sim p(s,a), s_f^- \sim p(s_f) \\ s_f^+ \sim p^{\pi(\cdot|\cdot)}(s_{t+}|s_t,a_t)}} \left[\mathcal{L}(s, a, s_f^+, s_f^-)\right],$$

$$\text{where} \quad \mathcal{L}(s, a, s_f^+, s_f^-) \triangleq \log \sigma(\underbrace{f(s, a, s_f^+)}_{\phi(s,a)^T\psi(s_f^+)}) + \log(1 - \sigma(\underbrace{f(s, a, s_f^-)}_{\phi(s,a)^T\psi(s_f^-)})). \tag{6}$$

Intuitively, the critic function $f(u = (s_t, a_t), v = s_f)$ now tells us the correlation between the current state-action pair and future outcomes, analogous to a Q-function. We therefore can use the critic function in the same way as actor-critic RL algorithms [66], figuring out which actions lead to the desired outcome. Because the Bayes-optimal critic function is a function of the state occupancy measure [84], $f^*(s, a, s_g) = \log\left(\frac{p^{\pi(\cdot|\cdot)}(s_{t+}=s_g|s,a)}{p(s_g)}\right)$, it can be used to express the Q-function:

**Lemma 4.1.** *The critic function that optimizes Eq. 6 is a Q-function for the goal-conditioned reward function (Eq. 1), up to a multiplicative constant $\frac{1}{p(s_f)}$: $\exp(f^*(s, a, s_f)) = \frac{1}{p(s_f)} \cdot Q_{s_f}^{\pi(\cdot|\cdot)}(s, a)$.*

The critic function can be viewed as an unnormalized density model, where $p(s_g)$ is the partition function. Much of the appeal of contrastive learning is it avoids estimating the partition function [46], which can be challenging; in the RL setting, it will turn out that this constant can be ignored when selecting actions. Our experiments show that learning a normalized density model works well when $s_g$ is low-dimensional, but struggles to solve higher-dimensional tasks.

This lemma relates the critic function to $Q_{s_f}^{\pi(\cdot|\cdot)}(s, a)$, not $Q_{s_f}^{\pi(\cdot|\cdot,s_f)}(s, a)$. The underlying reason is that the critic function combines together experience collected when commanding different goals. Prior goal-conditioned behavioral cloning methods [22, 41, 83, 120] perform similar sharing, but do not analyze the relationship between the learned policies and Q functions. Sec. 4.5 shows that this critic function can be used as the basis for a convergent RL algorithm under some assumptions.

### 4.3 Learning the Goal-Conditioned Policy

The learned critic function not only tells us the likelihood of future states, but also tells us how different actions change the likelihood of a state occurring in the future. Thus, to learn a policy for reaching a goal state, we choose the actions that make that state most likely to occur in the future:

$$\max_{\pi(a|s,s_g)} \mathbb{E}_{\pi(a|s,s_g)p(s)p(s_g)} \left[f(s, a, s_f = s_g)\right] \approx \mathbb{E}_{\pi(a|s,s_g)p(s)p(s_g)} \left[\log Q_{s_g}^{\pi(\cdot|\cdot)}(s, a) - \log p(s_g)\right]. \tag{7}$$

The approximation above reflects errors in learning the optimal critic, and will allow us to prove that this policy loss corresponds to policy improvement in Sec. 4.5, under some assumptions.

In practice, we parametrize the goal-conditioned policy as a neural network that takes as input the state and goal and outputs a distribution over actions. The actor loss (Eq. 7) is computed by sampling states and *random* goals from the replay buffer, sampling actions from the policy, and then taking gradients on the policy using a reparametrization gradient. On tasks with image observations, we add an action entropy term to the policy objective.

### 4.4 A Complete Goal-Conditioned RL Algorithm

The complete algorithm alternates between fitting the critic function using contrastive learning, updating the policy using Eq. 7, and collecting more data. Alg. 1 provides a JAX [13] implementation of the actor and critic losses. Note that the critic is parameterized as an inner product between a representation of the state-action pair, and a representation of the goal state: $f(s, a, s_g) = \phi(s, a)^T\psi(s_g)$. This parameterization allows for efficient computation, as we can compute the goal representations just once, and use them both in the positive pairs and the negative pairs. While this is common practice in representation learning, it is not exploited by most goal-conditioned RL algorithms. We refer to this method as `contrastive RL (NCE)`. In Appendix C, we derive a variant of this method (`contrastive RL (CPC)`) that uses the infoNCE bound on mutual information.

**Algorithm 1 Contrastive RL (NCE)**: the actor and critic losses for our method.

```
from jax.numpy import einsum, eye
from optax import sigmoid_binary_cross_entropy
def critic_loss(states, actions, future_states):
  sa_repr = sa_encoder(states, actions)   # (batch_dim, repr_dim)
  g_repr = g_encoder(future_states)        # (batch_dim, repr_dim)
  logits = einsum('ik,jk->ij', sa_repr, g_repr)  # <sa_repr[i], g_repr[j]> for all i,j
  return sigmoid_binary_cross_entropy(logits=logits, labels=eye(batch_size))

def actor_loss(states, goals):
  actions = policy.sample(states, goal=goals)  # (batch_size, action_dim)
  sa_repr = sa_encoder(states, actions)         # (batch_dim, repr_dim)
  g_repr = g_encoder(goals)                     # (batch_dim, repr_dim)
  logits = einsum('ik,ik->i', sa_repr, g_repr)  # <sa_repr[i], g_repr[i]>
  return -1.0 * logits
```

Contrastive RL (NCE) is an on-policy algorithm because it only estimates the Q-function for the policy that collected the data. However, in practice, we take as many gradient steps on each transition as standard off-policy RL algorithms [40, 48]. Please see Appendix E for full implementation details. We will also release an efficient implementation based on ACME [57] and JAX [13]. On a single TPUv2, training proceeds at $1100 \frac{\text{batches}}{\text{sec}}$ for state-based tasks and $105 \frac{\text{batches}}{\text{sec}}$ for image-based tasks; for comparison, our implementation of DrQ on the same hardware setup runs at $28 \frac{\text{batches}}{\text{sec}}$ ($3.9\times$ slower).[6] Architectures and hyperparameters are described in Appendix E.[7]

### 4.5 Convergence Guarantees

In general, providing convergence guarantees for methods that perform relabeling is challenging. Most prior work offers no guarantees [6, 22, 23] or guarantees under only restrictive assumptions [41, 120].

To prove that contrastive RL converges, we will introduce an additional filtering step into the method, throwing away some training examples. Precisely, we exclude training examples $(s, a, s_f)$ if the probability of the corresponding trajectory $\tau_{i:j} = (s_i, a_i, s_{i+1}, a_{i+1}, \cdots, s_j, a_j)$ sampled from $\pi(\tau \mid s_g)$ under the commanded goal $s_g$ is very different from the trajectory's probability under the actually-reached goal $s_j$:

$$\text{EXCLUDETRAJ}(\tau_{i:j}) = \delta \left( \left| \frac{\pi(\tau_{i:j} \mid s_g)}{\pi(\tau_{i:j} \mid s_j)} - 1 \right| > \epsilon \right).$$

While this modification is necessary to prove convergence, ablation experiments in Appendix Fig. 13 show that the filtering step can actually hurt performance in practice, so we do not include this filtering step in the experiments in the main text. We can now prove that contrastive RL performs approximate policy improvement.

**Lemma 4.2** (Approximate policy improvement). *Assume that states and actions are tabular and assume that the critic is Bayes-optimal. Let $\pi'(a \mid s, s_g)$ be the goal-conditioned policy obtained after one iteration of contrastive RL with a filtering parameter of $\epsilon$. Then this policy achieves higher rewards than the initial goal-conditioned policy:*

$$\mathbb{E}_{\pi'(\tau|s_g)}\left[\sum_{t=0}^{\infty} \gamma^t r_{s_g}(s_t, a_t)\right] \geq \mathbb{E}_{\pi(\tau|s_g)}\left[\sum_{t=0}^{\infty} \gamma^t r_{s_g}(s_t, a_t)\right] - \frac{2\gamma\epsilon}{1-\gamma} \quad \text{for all goals } s_g \in \{s_g \mid p_g(s_g) > 0\}.$$

The proof is in Appendix B. This result shows that performing contrastive RL on static dataset results in one step of approximate policy improvement. Re-collecting data and then applying contrastive RL over and over again corresponds to approximate policy improvement (see [10, Lemma 6.2]).

In summary, we have shown that applying contrastive learning to a particular choice of inputs results in an RL algorithm, one that learns a Q-function and (under some assumptions) converges to the reward-maximizing policy. Contrastive RL (NCE) is simple: it does not require multiple Q-values [40], target Q networks [88], data augmentation [69, 73], or auxiliary objectives [116, 137].

---

[6]The more recent DrQ-v2 [136] uses on 1 NVIDIA V100 GPU to achieve a training speed of $96/4 = 24 \frac{\text{batches}}{\text{sec}}$. The factor of 4 comes from an action repeat of 2 and an update interval of 2.

[7]Code and more results are available: `https://ben-eysenbach.github.io/contrastive_rl`

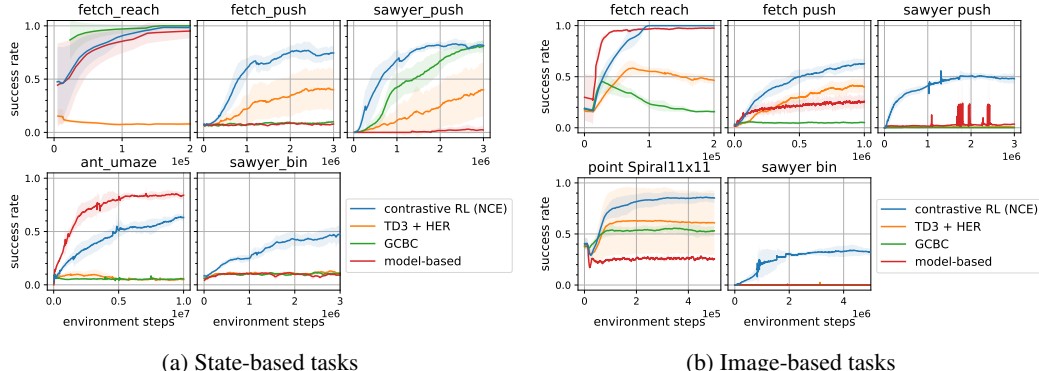

| (a) State-based tasks | (b) Image-based tasks |

Figure 2: **Goal-conditioned RL.** Contrastive RL (NCE) outperforms prior methods on most tasks. **Baselines**: `HER` [80] is a prototypical actor-critic method that uses hindsight relabeling [6]; `Goal-conditioned behavioral cloning (GCBC)` [22, 41, 83, 117] performs behavior cloning on relabeled experience; `model-based` fits a density model to the discounted state occupancy measure, similar on [21, 23, 60].

### 4.6 C-learning as Contrastive Learning

C-learning [29] is a special case of contrastive RL: it learns a critic function to distinguish future goals from random goals. Compared with contrastive RL (NCE), C-learning learns the classifier using temporal difference learning.[8] Viewing C-learning as a special case of contrastive RL suggests that contrastive RL algorithms might be implemented in a variety of different ways, each with relative merits. For example, `contrastive RL (NCE)` is much simpler than C-learning and tends to perform a bit better. Appendix D introduces another member of the contrastive RL family (`contrastive RL (NCE + C-learning)`) that tends to yield the best performance .

## 5 Experiments

Our experiments use goal-conditioned RL problems to compare contrastive RL algorithms to prior non-contrastive methods, including those that use data augmentation and auxiliary objectives. We then compare different members of the contrastive RL family, and show how contrastive RL can be effectively applied to the offline RL setting. Appendices E, F, and G contain experiments, visualizations, and failed experiments.

### 5.1 Comparing to prior goal-conditioned RL methods

**Baselines.** We compare three baselines. "HER" [80] is a goal-conditioned RL method that uses hindsight relabeling [6] with a high-performance actor-critic algorithm (TD3). This baseline is representative of a large class of prior work that uses hindsight relabeling [6, 76, 102, 106]. Like contrastive RL, this baseline does not assume access to a reward function. The second baseline is

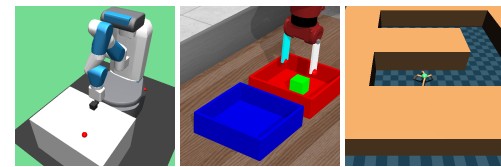

Figure 3: **Environments.** We show a subset of the goal-conditioned environments used in our experiments.

goal-conditioned behavioral cloning ("GCBC") [16, 22, 25, 41, 83, 96, 117, 120], which trains a policy to reach goal $s_g$ by performing behavioral cloning on trajectories that reach state $s_g$. GCBC is a simple method that achieves excellent results [16, 25] and has the same inputs as our method ($(s, a, s_f)$ triplets). A third baseline is a model-based approach that fits a density model to the future state distribution $p^{\pi(\cdot|\cdot)}(s_{t+} \mid s, a)$ and trains a goal-conditioned policy to maximize the probability of the commanded goal. This baseline is similar to successor representations [21] and prior multi-step models [23, 60]. Both contrastive RL (Alg. 1) and this model-based approach encode the future state distribution, but the output dimension of this model-based method depends on the state dimension. We, therefore, expect this approach to excel in low-dimensional settings but struggle with image-based tasks. Where possible, we use the same hyperparameters for all methods. We will

---

[8]The objectives are subtly different: C-learning estimates the probability that policy $\pi(\cdot \mid \cdot, s_g)$ visits state $s_f = s_g$, whereas contrastive RL (NCE) estimates the probability that any of the goal conditioned policies visit state $s_f$.

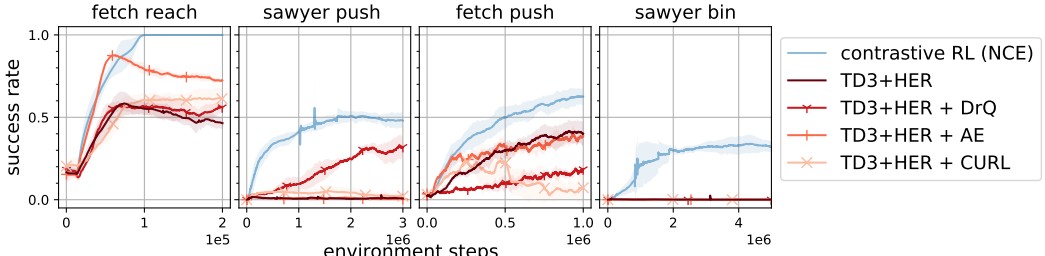

Figure 4: **Representation learning for image-based tasks.** While adding data augmentation and auxiliary representation objectives can boost the performance of the `TD3+HER` baseline, replacing the underlying goal-conditioned RL algorithm with one that resembles contrastive representation learning (i.e., ours) yields a larger increase in success rates. **Baselines**: `DrQ` [69] augments images and averages the Q-values across 4 augmentations; `auto encoder (AE)` adds an auxiliary reconstruction loss [32, 91, 93, 137]; `CURL` [116] applies RL on top of representations learned via augmentation-based contrastive learning.

include additional representation learning baselines when studying representations in the subsequent section.

**Tasks.** We compare it to a suite of goal-conditioned tasks, mostly taken from prior work. Four standard manipulation tasks include `fetch reach` and `fetch push` from Plappert et al. [97] and `sawyer push` and `sawyer bin` from Yu et al. [139]. We evaluate these tasks both with state-based observations and (unlike most prior work) image-based observations. The `sawyer bin` task poses an exploration challenge, as the agent must learn to pick up an object from one bin and place it at a goal location in another bin; the agent does not receive any reward shaping or demonstrations. We include two navigation tasks: `point Spiral11x11` is a 2D maze task with image observations and `ant umaze` [36] is a 111-dimensional locomotion task that presents a challenging low-level control problem. Where possible, we use the same initial state distribution, goal distribution, observations, and definition of success as prior work. Goals have the same dimension as the states, with one exception: on the `ant umaze` task, we used the global $XY$ position as the goal. We illustrate three of the tasks to the right. The agent does not have access to any ground truth reward function.

We report results in Fig. 2, using five random seeds for each experiment and plotting the mean and standard deviation across those random seeds. On the state-based tasks (Fig. 2a), most methods solve the easiest task (`fetch reach`) while only our method solves the most challenging task (`sawyer bin`). Our method also outperforms all prior methods on the two pushing tasks. The model-based baseline performs best on the `ant umaze` task, likely because learning a model is relatively easy when the goal is lower-dimensional (just the $XY$ location). On the image-based tasks (Fig. 2b), most methods make progress on the two easiest tasks (`fetch reach` and `point Spiral11x11`); our method outperforms the baselines on the three more challenging tasks. Of particular note is the success on `sawyer push` and `sawyer bin`: while the success rate of our method remains below 50%, no baselines make any progress on learning these tasks. These results suggest that contrastive RL (NCE) is a competitive goal-conditioned RL algorithm.

## 5.2 Comparing to prior representation learning methods

We hypothesize that contrastive RL may automatically learn good representations. To test this hypothesis, we compare contrastive RL (NCE) to techniques proposed by prior work for representation learning. These include data augmentation [69, 73, 136] ("DrQ") and auxiliary objectives based on an autoencoder [32, 91, 93, 137] ("AE") and a contrastive learning objective ("CURL") that generates positive examples using data augmentation, similar to prior work [89, 116, 118]. Because prior work has demonstrated these techniques in combination with actor-critic RL algorithms, we will use these techniques in combination with the actor-critic baseline from the previous section ("TD3 + HER"). While contrastive RL (NCE) resembles a contrastive representation learning method, it does not include any data augmentation or auxiliary representation learning objectives.

We show results in Fig. 4, with error bars again showing the mean and standard deviation across 5 random seeds. While adding the autoencoder improves the baseline on the `fetch reach` and adding DrQ improves the baseline on the `sawyer push`, contrastive RL (NCE) outperforms the prior methods on all tasks. Unlike these methods, contrastive RL does not use auxiliary objectives or additional domain knowledge in the form of image-appropriate data augmentations. These experiments do not show that representation learning is never useful, and do not show that contrastive

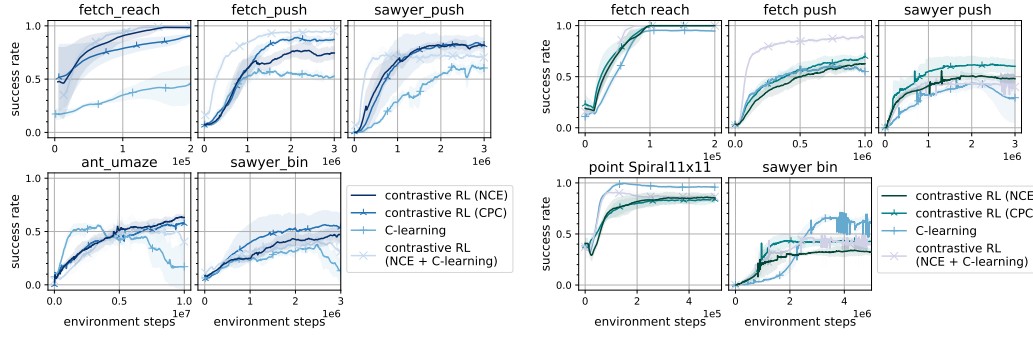

(a) state-based observations                    (b) image-based observations

Figure 5: **Contrastive RL design decisions.** Generalizing C-learning to a family of contrastive RL algorithms allowed us to identify algorithms that are much simpler (contrastive RL (NCE)) and that consistently achieve higher performance (contrastive RL (NCE + C-learning)).

RL cannot be improved with additional representation learning machinery. Rather, they show that designing RL algorithms that structurally resemble contrastive representation learning yields bigger improvements than simply adding representation learning tricks on top of existing RL algorithms.

### 5.3 Probing the dimensions of contrastive RL

Up to now, we have focused on the specific instantiation of contrastive RL spelled out in Alg. 1. However, there is a whole family of RL algorithms with contrastive characteristics. C-learning is a contrastive RL algorithm that uses temporal difference learning (Sec. 4.6). Contrastive RL (CPC) is a variant of Alg. 1 based on the infoNCE objective [95] that we derive in Appendix C Contrastive RL (NCE + C-learning) is a variant that combines C-learning with Alg. D (see Appendix D.). The aim of these experiments are to study whether generalizing C-learning to a family of contrastive RL algorithms was useful: do the simpler methods achieve similar performance, and do other methods achieve better performance?

We present results in Fig. 5, again plotting the mean and standard deviation across five random seeds. Contrastive RL (CPC) outperforms contrastive RL (NCE) on three, suggesting that swapping one mutual information estimator for another can sometimes improve performance, though both estimators can be effective. C-learning outperforms contrastive RL (NCE) on three tasks but performs worse on other tasks. Contrastive RL (NCE + C-learning) consistently ranks among the best methods. These experiments demonstrate that the prior contrastive RL method, C-learning [29], achieves good results on most tasks; generalizing C-learning to a family of contrastive RL algorithms resulting in new algorithms that achieve higher performance and can be much simpler.

### 5.4 Partial Observability and Moving Cameras

Many realistic robotics tasks exhibit partial observability, and have cameras that are not fixed but rather attached to moving robot parts. Our next experiment tests if contrastive RL can cope with these sorts of challenges. To study this question, we modified the `sawyer push` task so that the camera tracks the hand at a fixed distance, as if it were rigidly mounted to the arm. This means that, at the start of the episode, the scene is occluded by the wall at the edge of the table, so the agent cannot see the location of the puck (see Fig. 6 (left)). Nonetheless, contrastive RL

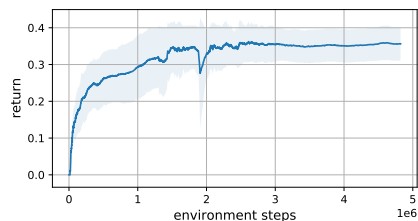

Figure 6: **Partial observability and moving cameras.** Contrastive RL can solve partially observed tasks.

(NCE) successfully handles this partial observability, achieving a success rate of around 35%. Fig. 6 (left) shows an example rollout and Fig. 6 (right) shows the learning curve. For comparison, the success rate when using the fixed static camera was 75%. Taken together, these results suggest that contrastive RL can cope with moving cameras and partial observability, while also suggesting that improved strategies (e.g., non-Markovian architectures) might achieve even better results.

Table 1: **Offline RL on D4RL AntMaze** [36]. Contrastive RL outperforms all baselines in 5 out of 6 tasks.

| | no TD | | | | | uses TD | |
|---|---|---|---|---|---|---|---|
| | BC | DT | GCBC | Contrastive RL + BC | | TD3+BC* | IQL* |
| | | | | 2 nets | 5 nets | | |
| umaze-v2 | 54.6 | 65.6 | 65.4 | 81.9 ($\pm$1.7) | 79.8 ($\pm$1.4) | 78.6 | **87.5** |
| umaze-diverse-v2 | 45.6 | 51.2 | 60.9 | **75.4** ($\pm$3.5) | **77.6** ($\pm$2.8) | 71.4 | 62.2 |
| medium-play-v2 | 0.0 | 1.0 | 58.1 | **71.5** ($\pm$5.2) | 72.6 ($\pm$2.9) | 10.6 | **71.2** |
| medium-diverse-v2 | 0.0 | 0.6 | 67.3 | **72.5** ($\pm$2.8) | 71.5 ($\pm$1.3) | 3.0 | **70.0** |
| large-play-v2 | 0.0 | 0.0 | 32.4 | 41.6 ($\pm$6.0) | **48.6** ($\pm$4.4) | 0.2 | 39.6 |
| large-diverse-v2 | 0.0 | 0.2 | 36.9 | **49.3** ($\pm$6.3) | **54.1** ($\pm$5.5) | 0.0 | 47.5 |

\* While TD3+BC and IQL report results on the `-v0` tasks, the change to `-v2` has a negligible effect on TD methods [8].

## 5.5 Contrastive RL for Offline RL

Our final experiment studies whether the benefits from contrastive RL (NCE) transfer to the offline RL setting, where the agent is prohibited from interacting with the environment. We use the benchmark AntMaze tasks from the D4RL benchmark [36], as these are goal-conditioned tasks commonly studied in the offline setting.

We adapt contrastive RL (NCE) to the offline setting by adding an additional (goal-conditioned) behavioral cloning term to the policy objective (Eq. 7), using a coefficient of $\lambda$:

$$\max_{\pi(a|s, s_g)} \mathbb{E}_{\pi(a|s,s_g)p(s,a_{\text{orig}},s_g)} \left[ (1 - \lambda) \cdot f(s, a, s_f = s_g) + \lambda \cdot \log \pi(a_{\text{orig}} \mid s, s_g) \right].$$

Note that setting $\lambda = 1$ corresponds to GCBC [16, 22, 25, 41, 83, 96, 117, 120], which we will include as a baseline. Following TD3+BC [39], we learn multiple critic functions (2 and 5) and take the minimum when computing the actor update. We also compare to prior offline RL methods that eschew TD learning: (unconditional) behavioral cloning (BC), the implementation of GCBC from [25] (which refers to GCBC as RvS-G), and a recent method based on the transformer architecture (DT [16]). Lastly, we compare with two more complex methods that use TD learning: TD3+BC [39] and IQL [68]. Unlike contrastive RL and GCBC, these TD learning methods do not perform goal relabeling. We use the numbers reported for these baselines in prior work [25, 68].

As shown in Table 1, contrastive RL (NCE) outperforms all baselines on five of the six benchmark tasks. Of particular note are the most challenging "-large" tasks, where contrastive RL achieves a 7% to 9% absolute improvement over IQL. We note that IQL does not use goal relabeling, which is the bedrock of contrastive RL. Compared to baselines that do not use TD learning, the benefits are more pronounced, with a median (absolute) improvement over GCBC of 15%. The performance of contrastive RL improves when increasing the number of critics from 2 to 5, suggesting that the key to solving more challenging offline RL tasks may be increased capacity, rather than TD learning. Taken together, these results show the value of contrastive RL for offline goal-conditioned tasks.

## 6 Conclusion

In this paper, we showed how contrastive representation learning can be used for goal-conditioned RL. This connection not only lets us re-interpret a prior RL method as performing contrastive learning, but also suggests a family of contrastive RL methods, which includes simpler algorithms, as well as algorithms that attain better overall performance. While this paper might be construed to imply that RL is more or less important than representation learning [72, 75, 112, 114], we have a different takeaway: that it may be enough to build RL algorithms that *look like* representation learning.

One limitation of this work is that it looks only at the goal-conditioned RL problems. How these methods might be applied to arbitrary RL problems remains an open problem, though we note that recent algorithms for this setting [28] already bear a resemblance to contrastive RL. Whether the rich set of ideas from contrastive learning might be used to construct even better RL algorithms likewise remains an open question.

**Acknowledgements.** Thanks to Hubert Tsai, Martin Ma, and Simon Kornblith for discussions about contrastive learning. Thanks to Kamyar Ghasemipour, Suraj Nair, and anonymous reviewers for feedback on the paper. Thanks to Ofir Nachum, Daniel Zheng, and the JAX and Acme teams for helping to release and debug the code. This material is supported by the Fannie and John Hertz Foundation and the NSF GRFP (DGE1745016). UC Berkeley research is also supported by gifts from Alibaba, Amazon Web Services, Ant Financial, CapitalOne, Ericsson, Facebook, Futurewei, Google, Intel, Microsoft, Nvidia, Scotiabank, Splunk and VMware.

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
