# A    Additional Related Work

Our work is also related to unsupervised skill discovery [1, 19, 27, 42, 52, 74, 110], in that the algorithm learns multiple policies by interacting in the environment without a reward function. Both these skill learning algorithms and our contrastive algorithm optimize a lower bound on mutual information. Indeed, prior work has discussed the close connection between mutual information and goal-conditioned RL [19, 127]. The key challenge in making this connection is *grounding* the skills, so that each skill corresponds to a specific goal-conditioned policy.

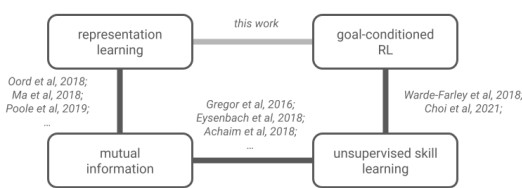

Figure 7: **Connecting related work.** This work helps draw connections between prior work, filling in a missing link.

While the skills can be grounded by manually-specifying the critic used for maximizing mutual information [19], manually-specifying the critic for high-dimensional tasks (e.g., images) would be challenging. Our work takes a different approach to grounding, one based on reasoning directly about continuous probabilities. In the end, our method will learn skills that each corresponds to a specific goal-conditioned policy and will be scalable to high-dimensional tasks.

Fig. 7 highlights some of the connections between related work. Prior work has thoroughly explained how many representation learning methods correspond to a lower bound on mutual information [84, 99]. Prior work in RL has proposed unsupervised skill learning algorithms using similar mutual information objectives [1, 27, 42], and more recent work has connected these unsupervised skills learning algorithms to goal-reaching. The key contribution of this paper is to connect representation learning to goal-conditioned RL.

# B    Proofs

## B.1    Q-function are equivalent to the discounted state occupancy measure

This section proves Proposition 1. We start by recalling the definition of the discounted state occupancy measure (Eq. 3):

$$p(s_{t+} = s_g) = (1 - \gamma) \sum_{t=0}^{\infty} \gamma^t p_t^{\pi(\cdot|\cdot,s_g)}(s_t = s_g). \tag{8}$$

We first analyze the term for $t = 0$, and then analyze the term for $t > 0$. The probability of visiting a state at time $t = 0$ is just the initial state distribution:

$$p_0^{\pi(\cdot|\cdot,s_g)}(s_t = s_g) = p_0(s_0 = s_g).$$

We can now rewrite Eq. 8 as

$$p(s_{t+} = s_g) = (1 - \gamma)p_0(s_0 = s_g) + (1 - \gamma) \sum_{t=1}^{\infty} \gamma^t p_t^{\pi(\cdot|\cdot,s_g)}(s_t = s_g). \tag{9}$$

For $t > 1$, we can write the term as follows:

$$p_t^{\pi(\cdot|\cdot,s_g)}(s_t = s_g) = \mathbb{E}_{p_{t-1}^{\pi(\cdot|\cdot,s_g)}(s_{t-1})\pi(a_{t-1}|s_{t-1},s_g)} \left[ p_t(s_t = s_g \mid s_{t-1}, a_{t-1}) \right]$$

$$= \mathbb{E}_{p_{t-1}^{\pi(\cdot|\cdot,s_g)}(s_{t-1}),\pi(a_{t-1}|s_{t-1},s_g)} \left[ p(s_t = s_g \mid s_{t-1}, a_{t-1}) \right]$$

$$= \mathbb{E}_{\tau \sim \pi(\tau|s_t)} \left[ p(s_t = s_g \mid s_{t-1}, a_{t-1}) \right].$$

In the second line, we have used the Markov property to say that the probability of visiting $s_g$ at time $t$ depends only on dynamics, $p(s_{t+1} \mid s_t, a_t)$. In the third line, we have rewritten the expectation over trajectories, using $s_{t-1}$ and $a_{t-1}$ and the $t - 1^{\text{th}}$ state-action pair in the trajectory. Substituting this

into Eq. 9, we get

$$p(s_{t+} = s_g) = (1-\gamma)p_0(s_0 = s_g) + (1-\gamma)\sum_{t=1}^{\infty}\gamma^t \mathbb{E}_{\tau \sim \pi(\tau|s_g)}\left[p(s_t = s_g \mid s_{t-1}, a_{t-1})\right]$$

$$= (1-\gamma)p_0(s_0 = s_g) + (1-\gamma)\sum_{t=0}^{\infty}\gamma^t \mathbb{E}_{\tau \sim \pi(\tau|s_g)}\left[p(s_{t+1} = s_g \mid s_t, a_t)\right]$$

$$= (1-\gamma)p_0(s_0 = s_g) + (1-\gamma)\mathbb{E}_{\tau \sim \pi(\tau|s_g)}\left[\sum_{t=0}^{\infty}\gamma^t p(s_{t+1} = s_g \mid s_t, a_t)\right]$$

$$= \mathbb{E}_{\tau \sim \pi(\tau|s_g)}\left[(1-\gamma)p_0(s_0 = s_g) + (1-\gamma)\sum_{t=0}^{\infty}\gamma^t p(s_{t+1} = s_g \mid s_t, a_t)\right]$$

$$= \mathbb{E}_{\tau \sim \pi(\tau|s_g)}\left[\sum_{t=0}^{\infty}\gamma^t r_g(s_t, a_t)\right].$$

On the second line, we have changed the bounds of the summation to start at 0, and changed the terms inside the summation accordingly. On the third line, we applied linearity of expectation to move the summation inside the expectation. On the fourth line, we applied linearity of expectation again to move the term for $t = 0$ inside the expectation. Finally, we substituted the definition of $r_g(s, a)$ to obtain the desired result.

### B.2 Contrastive RL is Policy Improvement

This section proves the Contrastive RL (NCE) corresponds to policy improvement, yielding policies with higher rewards at each iteration (Lemma 4.2).

*Proof.* The main idea of the proof is to relate the Q-values for the average policy to the Q-values for the goal-conditioned policy. We do this by employing the result from [31, Appendix C.2], where $\epsilon$ is the parameter for filtered relabeling (Sec. 4.5):

$$\left|Q^{\beta(\cdot mid \cdot, e)}(s, a, e) - Q^{\beta(\cdot|\cdot, e')}(s, a, e)\right| \leq \epsilon.$$

This result means that we are doing policy improvement with approximate Q-values. Then, [10, Lemma 6.1] tells that doing policy improvement using approximate Q-values gives us approximate policy improvement:

$$\mathbb{E}_{\pi'(\tau|s_g)}\left[\sum_{t=0}^{\infty}\gamma^t r_{s_g}(s_t, a_t)\right] \geq \mathbb{E}_{\pi(\tau|s_g)}\left[\sum_{t=0}^{\infty}\gamma^t r_{s_g}(s_t, a_t)\right] - \frac{2\gamma\epsilon}{1-\gamma} \qquad \text{for all goals } s_g \in \{s_g \mid p_g(s_g) > 0\}.$$

$\square$

## C Contrastive RL (CPC)

In this section, we derive a version of contrastive RL based on the infoNCE objective [95]. Compared with the NCE objective used in contrastive RL (NCE), this objective uses a categorical cross entropy loss instead of a binary cross entropy loss. We replace Eq. 6 with the following infoNCE objective [95]:

$$\max_f \mathbb{E}_{(s,a)\sim p(s,a), s_f^{(1)}\sim p^{\pi(\cdot|\cdot)}(s_{t+}|s,a) \atop s_f^{(2:B)}\sim p(s_f)}\left[\log p^{(1)}\right],$$

where $p^{(1)}$ is the first coordinate of the softmax over the critic:

$$p = \text{SOFTMAX}([f(s, a, s_f^{(1)}), \cdots, f(s, a, s_f^{(b)})]).$$

The optimal critic for the infoNCE loss satisfies [84, 95, 99]

$$f^*(s, a, s_f) = \log\left(\frac{p^{\pi(\cdot|\cdot)}(s_{t+} = s_f \mid s, a)}{p(s_f)c(s, a)}\right),$$

where $c(s,a)$ is an arbitrary function. Thus, there are many optimal critics. Choosing actions that maximize the critic $f^*$ does not necessarily correspond to choosing actions that maximize the probability of the future state. Thus, we need to regularize $c(s,a)$ so that it does not depend on $a$. We do this by introducing a regularizer, based on [123]:

$$\min_f E_{s_f^{(1:B)} \sim p(s_f)} \text{LOGSUMEXP}([f(s,a,s_f^{(1)}), \cdots, f(s,a,s_f^{(b)})])^2.$$

To provide some intuition for this regularizer, consider applying this regularizer to an optimal critic:

$$\text{LOGSUMEXP}([f^*(s,a,s_f^{(1)}), \cdots, f^*(s,a,s_f^{(b)})])^2$$

$$= \left( \log \frac{1}{c(s,a)} \sum_{s_f} \frac{p^{\pi(\cdot|\cdot)}(s_{t+} = s_f \mid s,a)}{p(s_f)c(s,a)} \right)^2$$

$$= \left( \log \sum_{s_f \in s_f^{(1:B)}} \frac{p^{\pi(\cdot|\cdot)}(s_{t+} = s_f \mid s,a)}{p(s_f)} - \log c(s,a) \right)^2$$

$$\approx \left( \log \sum_{s_f \in s_f^{(2:B)}} \frac{p^{\pi(\cdot|\cdot)}(s_{t+} = s_f \mid s,a)}{p(s_f)} - \log c(s,a) \right)^2$$

$$\approx \left( \log \mathbb{E}_{s_f \sim p(s_f)} \left[ \frac{p^{\pi(\cdot|\cdot)}(s_{t+} = s_f \mid s,a)}{p(s_f)} \right] - \log c(s,a) \right)^2$$

$$= \left( - \log c(s,a) \right)^2 .$$

In the third line we ignore the positive term; this is reasonable if the batch size is large enough. In the third line we replaced the sum with an expectation; this is biased because $\log(\cdot)$ is not a linear function. Thus, this regularizer (approximately) regularizes $c(s,a)$ to be close to 1 for all states and actions. By reducing the dependency of $c(s,a)$ on the actions $a$, we can ensure that actions that maximize the critic do maximize the probability of reaching the desired goal. In practice, we add this regularizer with the infoNCE objective, using a coefficient of 1e-2 on the regularizer.

## D   Contrastive RL (NCE + C-learning)

In this section we describe `contrastive RL (NCE + C-learning)` the combined NCE + C-learning method used in Sec. 5.3 (Fig. 5). Mathematically, the NCE + C-learning objective is a simple, unweighted sum of the C-learning objective and the NCE objective:

$$\mathcal{L}(f) = (1-\gamma)\mathbb{E}_{(s,a)\sim p(s,a), s_f^+ \sim p(s_{t+1}|s_t,a_t)}[\log \sigma(f(s,a,s_f^+))]$$

$$+ \gamma \mathbb{E}_{\substack{s_g \sim p_g(s_g), (s_t,a_t) \sim p(s,a), \\ s_{t+1} \sim p(s_{t+1}|s_t,a_t), a_{t+1} \sim \pi(a_{t+1}|s_{t+1},s_g)}} \left[ \underbrace{\frac{p(s_{t+} = s_g \mid s_t,a_t)}{p(s_f = s_g)} \log \sigma(f(s,a,s_f = s_g))}_{\approx \exp(f(s_{t+1},a_{t+1},s_g))} \right]$$

$$+ \mathbb{E}_{s_g \sim p_g(s_g), (s,a)\sim p(s,a)} [\log(1 - \sigma(f(s,a,s_g)))]$$

$$+ \mathbb{E}_{(s,a)\sim p(s,a), s_f^+ \sim p(s_{t+}|s_t,a_t)}[\log \sigma(f(s,a,s_f^+))] + \mathbb{E}_{(s,a)\sim p(s,a), s_f^- \sim p(s_f)}[\log(1 - \sigma(f(s,a,s_f^-)))].$$

While we could use half the batch to compute each of the loss terms, we can increase the effective sample size by being careful with how the terms are estimated. First, we note that the first two terms of each loss are similar – sample a future state (either the next state or a future state) and label it as a positive. We can thus combine these two terms by sampling from a mixture of these two distributions,

$$\tilde{p}(s_f \mid s_t,a_t) = \frac{1-\gamma}{1 + 1 - \gamma} p(s_{t+1} = s_f \mid s_t,a_t) + \frac{1}{1+1-\gamma} p(s_{t+} = s_f \mid s_t,a_t),$$

and scaling the resulting loss by $1 + 1 - \gamma = 2 - \gamma$:

$$\mathcal{L}_1(f) \triangleq (1-\gamma)\mathbb{E}_{(s,a)\sim p(s,a), s_f^+ \sim p(s_{t+1}|s_t,a_t)}[\log \sigma(f(s,a,s_f^+))] + \mathbb{E}_{(s,a)\sim p(s,a), s_f^+ \sim p(s_{t+}|s_t,a_t)}[\log \sigma(f(s,a,s_f^+))]$$

$$= (2-\gamma)\mathbb{E}_{(s,a)\sim p(s,a), s_f^+ \sim \tilde{p}(s_f|s_t,a_t)}[\log \sigma(f(s,a,s_f^+))]$$

This trick increases the effective sample size by 96% ($130 \rightarrow 256$, as measured using [64]).

Both losses also contain terms that are an expectation over random goals. We can likewise combine those terms:

$$\mathcal{L}_2(f) \triangleq \gamma \mathbb{E}_{\substack{s_g \sim p_g(s_g),(s_t,a_t) \sim p(s,a), \\ s_{t+1} \sim p(s_{t+1}|s_t,a_t),a_{t+1} \sim \pi(a_{t+1}|s_{t+1},s_g)}} \left[ \lfloor \exp(f(s_{t+1},a_{t+1},s_g)) \rfloor_{\text{sg}} \log \sigma(f(s,a,s_f = s_g)) \right]$$

$$+ \mathbb{E}_{s_g \sim p_g(s_g),(s,a) \sim p(s,a)} \left[ \log(1 - \sigma(f(s,a,s_g))) \right] + \mathbb{E}_{(s,a) \sim p(s,a),s_f^- \sim p(s_f)} [\log(1 - \sigma(f(s,a,s_f^-)))]$$

$$= \gamma \mathbb{E}_{\substack{s_g \sim p_g(s_g),(s_t,a_t) \sim p(s,a), \\ s_{t+1} \sim p(s_{t+1}|s_t,a_t),a_{t+1} \sim \pi(a_{t+1}|s_{t+1},s_g)}} \left[ \lfloor \exp(f(s_{t+1},a_{t+1},s_g)) \rfloor_{\text{sg}} \log \sigma(f(s,a,s_f = s_g)) \right]$$

$$+ 2\mathbb{E}_{s_g \sim p_g(s_g),(s,a) \sim p(s,a)} \left[ \log(1 - \sigma(f(s,a,s_g))) \right].$$

Note that estimating the first term in $\mathcal{L}_2$ requires sampling an action for each next state and goal pair. This prohibits us from using the same outer product trick as in Sec. 4.4 to estimate this term. While we could still use that trick to estimate the second term in $\mathcal{L}_2$, we found that doing so hurt performance. We hypothesize that the reason is that this creates an imbalance in the gradients – some goals are labeled as negatives but are not also labeled as positives. Thus, we do not use the outer product trick for this method. The final objective is $\mathcal{L}(f) = \mathcal{L}_1(f) + \mathcal{L}_2(f)$.

# E Experimental Details

We implemented contrastive RL and the baselines using the ACME RL library [57] in combination with JAX [13]. Precisely, we took the SAC agent[9] and made the modifications below. Unless otherwise mentioned, we used the same hyperparameters as this implementation.

1. Implemented environments that returned observations that contained the original observation concatenated with the goal. While the goals are resampled when sampling from the replay buffer, assuming that observations include the goals means that the Q-function and policy networks do not need to include an additional input for the goal.

2. Modified the replay buffer to use trajectories rather than transitions. This allows us to sample $(s_t, a_t, s_f)$ triplets.

3. Modified the critic network to be parametrized as an inner product between state-action representations and goal representations.

4. Modified the critic loss based on Alg. 1. Note that the actor loss does not need to be modified (except for removing the entropy term for state-based tasks).

We summarize the hyperparameters in Table 2. Both the state-action encoder and the goal encoder are fully-connected neural networks with 2 layers of size 256 with ReLU activations. We found that normalizing the final representation, applying a final activation function, or using a learnable temperature hurt performance, so we do not use these tricks. For image-based tasks, observations have size $(64, 64, 3)$, and we use a CNN encoder from prior work [88] to encode the observations before passing them to the encoders. The policy has a similar architecture: an image-encoder for image-based tasks, followed by 2 fully connected layers with size 256 and ReLU activations.

Table 2: Hyperparameters for our method and the baselines.

| hyperparameter | value |
|---|---|
| batch size | 256 |
| learning rate | 3e-4 for all components |
| discount | 0.99 |
| actor target entropy | 0 for state-based experiments, $-\dim(a)$ for image-based experiments |
| target EMA term (for TD3 and SAC) | 0.005 |
| image encoder architecture | Taken from Mnih et al. [88] |
| image decoder architecture (for auto-encoder and model-based baselines) | Taken from Ha and Schmidhuber [47] |
| hidden layers sizes (for actor and representations) | (256, 256) |
| initial random data collection | 10,000 transitions |
| replay buffer size | 1,000,000 transitions |
| samples per insert[1] | 256 |
| train-collect interval[2] | 16 for state-based tasks, 64 for image-based tasks |
| representation dimension $(\dim(\phi(s,a)), \dim(\psi(s_g)))$ | 64 |
| actor minimum std dev | 1e-6 |
| number of augmentations (for DrQ only) | 4 |
| logsumexp regularizer coefficient (for CPC only) | 1e-2 |
| action repeat | None |
| goals for actor loss | random states (not future states) |

[1] How many times is each transition used for training before being discarded.

[2] We collect $N$ transitions, add them to the buffer, and then do $N$ gradient steps using the experience sampled randomly from the buffer.

---

[9] https://github.com/deepmind/acme/tree/master/acme/agents/jax/sac

Table 3: Changes to hyperparameters for offline RL experiments (Fig. 1).

| hyperparameter | value |
|---|---|
| batch size | $256 \rightarrow 1024$ |
| representation dimension | $64 \rightarrow 16$ |
| hidden layers sizes (for actor and representations) | $(256, 256) \rightarrow (1024, 1024)$, as in [25]. |
| goals for actor loss | future states |

## E.1 Environments

**fetch reach** (image, state) – This task is taken from Plappert et al. [97]. This task involves moving a robotic arm to a goal location in free space. The benchmark specifies success as reaching within 5cm of the goal.

**fetch push** (image, state) – This task is taken from Plappert et al. [97]. This task involves using a robotic arm to push an object across a table to a goal location. The benchmark specifies success as reaching within 5cm of the goal.

**sawyer push** (image, state) – This task is taken from Yu et al. [139]. This task involves using a robotic arm to push an object across table to a goal location. The benchmark specifies success as reaching within 5cm of the goal.

**ant umaze** (state) – This task is taken from Fu et al. [36]. This task involves controlling an ant-like robot towards a goal location, which is sampled randomly in a "U"-shaped maze. Unlike all other tasks in this paper, the goal was lower dimensional than the observation: the goal was just the XY coordinate of the desired position. Following prior work [15, 115], success is defined as reaching within 0.5m of the goal.

**sawyer bin** (image, state) – This task is taken from Yu et al. [139]. This task involves using a robotic arm to pick up a block from one bin and place it at a goal location in another bin. The benchmark specifies success as reaching within 5cm of the goal.

**point Spiral11x11** (image) – This task is an image-based version of the 2D navigation tasks in Eysenbach et al. [30]. This task involves directly controlling the XY coordinates of an agent to reach a goal in a spiral-shaped maze (see Fig. 16). We define success as reaching within 2m of the goal.

# F  Additional Experiments

## F.1  Linear regression with the learned features

To study the learned representations in isolation we take the state-action representations $\phi(s, a)$ trained on the image-based `point NineRooms` task, and run a linear probe [3, 51] experiment to see whether the representations have learned to encode task-relevant information (the shortest path distance to the goal).

We use the task of nine-room navigation and run Contrastive RL and TD3+HER on it. We visualize the environment in Fig. 8a and the agent randomly initialized in one of the nine rooms is commanded to go to the goal position. We dump the replay buffer during training as the dataset and run a linear regression to predict the shortest distance between the agent and the goal. Note that this shortest path distance is not the Euclidean distance since there are walls blocking the way. Fig. 8b shows that features learned by contrastive RL can predict this distance better than all baselines.

As shown in Fig. 8b, contrastive RL (NCE) learns representations that achieve lower test error than those learned by TD3+HER and by a random CNN encoder.

## F.2  When is contrastive learning better than learning a foreward model?

In Fig. 2a, we observed that the model-based baseline performed well on the `ant umaze` task, but poorly on many of the other tasks. One explanation is that the model-based approach will perform well when the goal is relatively low-dimensional, and that contrastive learning will be more useful in settings with higher-dimensional goals. We tested this experiment on the 7-dimensional `sawyer push` environment. We applied both contrastive RL and the model-based baseline to versions of this task where the goal was varied from 1-dimensional to 7-dimensional. Note that changing the goal dimension changes the task: a 1-dimensional goal

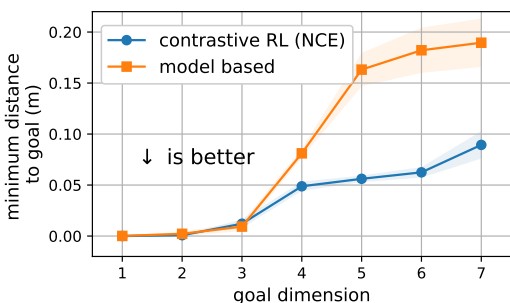

Figure 9: Contrasive learning outperforms a forward model when the goal is 4-dimensional or larger. Error bars show the standard deviation across 5 random seeds.

corresponds to moving the gripper to the correct X position, whereas a 7-dimensional goal corresponds to moving the object and gripper to the correct poses. We measured the Euclidean distance to the goal ($\downarrow$ is better). We show results in Fig. 9. As expected, higher-dimensional goals are a bit more challenging to achieve. What we are really interested in is the *gap* between the model-based approach and the contrastive RL, which opens up starting with a 4-dimensional goal. Altogether, this

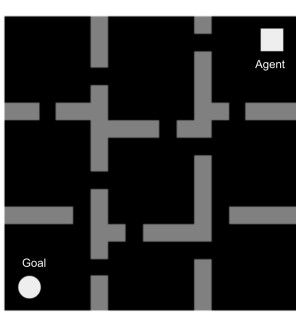

(a) **Nine-Room environment.**

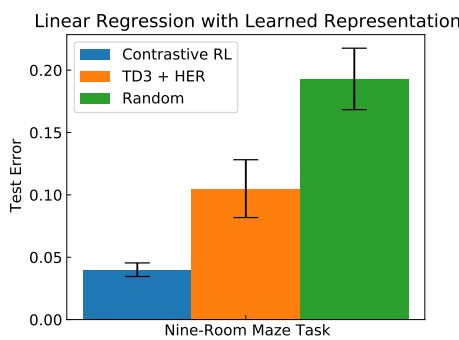

(b) **Linear probe experiment.**

Figure 8: **Linear regression with the learned features.** Contrastive RL can produce better features for predicting the shortest-path distance, indicating that the learned features have captured highly non-linear information about the environment dynamics.

experiment provides some evidence that contrastive RL may be preferred over a forward model, even for tasks with very low dimensional goals.

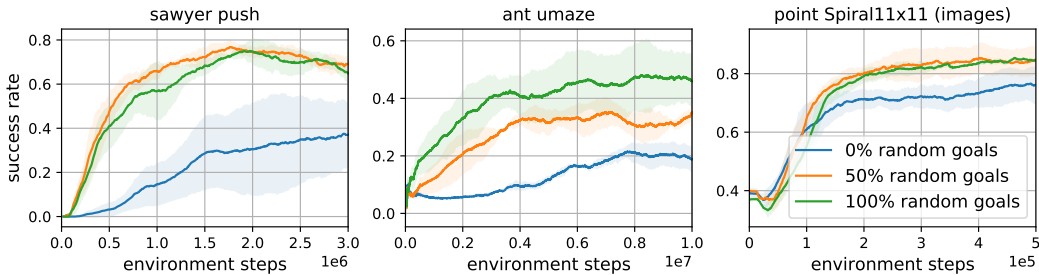

Figure 10: **Goals used for the actor loss.** Goals are either sampled from the distribution over future states or from a distribution of random states. Error bars show the standard deviation across 5 random seeds.

### F.3  Goals used in the actor loss

In theory, the distribution of goals for the actor loss (Eq. 7) does not affect the optimal policy, as long as the distribution has full support. In our experiments, we sampled these goals randomly, in the same way that we sampled negative examples for contrastive learning. We ran an ablation experiment to study this decision, and show results in Fig. 10. These results show that sampling future goals consistently performs poorly, perhaps because it results in only training the policy on how to reach "easy" goals. A mixture of future goals and random goals works much better, but the best results seem to come from training on only random goals.

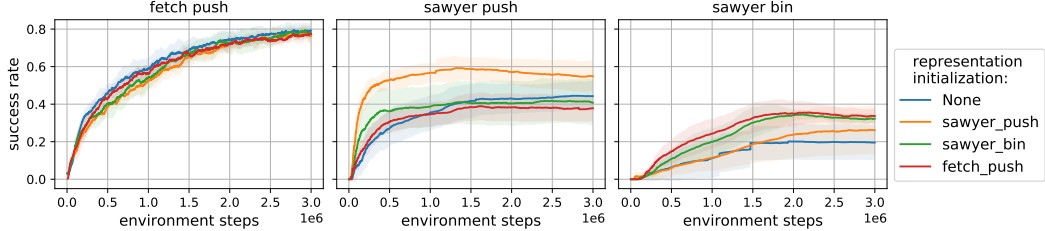

Figure 11: **Transferring representations to solve new tasks.** After training the representations on one task for 1M environment steps, we used them to initialize a new agent for solving a new task.

### F.4  Transferring representations to solve new tasks

In this experiment, we studied whether the representations learned by contrastive RL (NCE) for one task might be useful for solving another task. We started by training contrastive RL (NCE) on three image-based tasks: `fetch push`, `sawyer push`, and `sawyer bin`. The observations for all tasks look different and the sawyer and fetch tasks have different robots. The two sawyer tasks look the most similar because they both come from the metaworld [139] benchmark suite. We used the representations learned for each of these tasks to initialize a second contrastive RL agent, which we used to solve this same set of tasks. We were primarily interested in transfer – do the representations from one task help in learning to solve another task? Intuitively, even if the tasks are different, a good representation will capture some structural properties (e.g., identifying the robot arm, and identifying objects), which should transfer across the task.

We show results in Fig. 11. After training on the first task for 1M environment steps, we used the learned representation as initialization for solving the new task. On the `fetch push` task, we see little benefit from using pretrained representations, perhaps because the task is relatively easy. On the `sawyer push`, we see the largest benefit from pretraining the representations on the same task as the target task. More interestingly, we see a small benefit from taking the representations learned on the `sawyer bin` task and using those to solve the `sawyer push`. On the most challenging task, `sawyer bin`, using representations pretrained on either `fetch push` or `sawyer bin` can accelerate

the solving of this task. Taken together, these results suggest that transferring the representations from one task to another is sometimes useful.

## F.5   Robustness to Environment Perturbations

We ran an preliminary experiment to study whether the image-based policies learned by contrastive RL (NCE) were robust to perturbations in the environment. We took an agent trained on the `fetch push` with image-observations, and evaluated the agent on four variants of the environment (see Fig. 12):

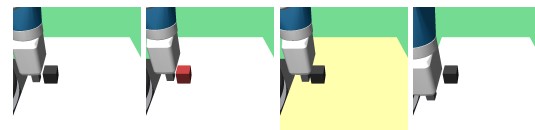

Figure 12: Perturbations to the image-based `fetch push` environment.

- Original environment, without modification;
- Object color changed from black to red;
- Table color changed from white to yellow;
- Initial arm position moved towards the camera.

In each setting, we evaluate the success rate over 20 trials, and repeated 5 times to compute standard deviations (for a total of 100 trials). The learned agent was robust to the object color, with the success rate changing from $78 \pm 5\%$ to $73 \pm 10\%$. The agent was also robust to the change in initial position ($87 \pm 6\%$). However, changing the table color caused the agent to fail ($0 \pm 0\%$), perhaps because the table color consumes a large fraction of the image pixels.

## F.6   Additional figures

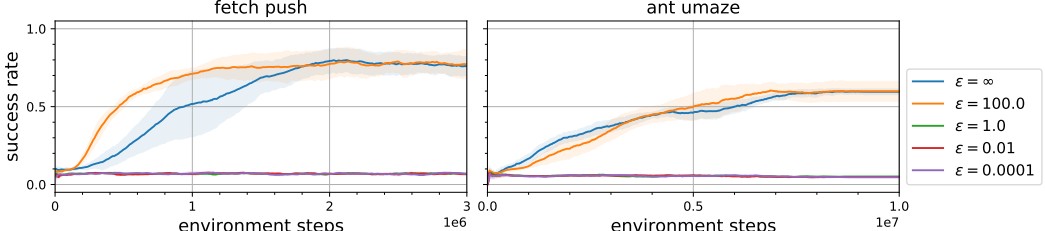

Figure 13: **Filtered relabeling.** We filter the relabeled experience so that the agent only trains on experience where the probability under the commanded goal is similar to the probability under the actually-reached goal. While such filtering is required to prove convergence, these results suggest that good performance can be achieved without this filtering step.

This section presents additional figures.

- Fig. 13 compares contrastive RL (NCE) with varying values of the filtering parameter $\epsilon$, described in Sec. 4.5.
- Fig. 14 – This plot shows a TSNE embedding of the state-action representations $\phi(s, a)$ for one trajectory of the `bin picking` task. This experiment uses image observations.
- Fig. 15 – This plot shows a TSNE embedding of the state-action representations from the same `bin picking` task. We sampled states and actions using a trained agent. After computing the TSNE embedding, we used RasterFairy [65] to rectify the embeddings to a grid.
- Fig. 16 – A TSNE embedding of image representations from the `point Spiral11x11` task.
- Fig. 17 – Using the same representations for the `point Spiral11x11` task, we measure the similarity between the critic gradients when evaluated at the same state but different goals, $\langle \frac{\partial f}{\partial s}\big|_{(s,g)}, \frac{\partial f}{\partial s}\big|_{(s,g')} \rangle$.

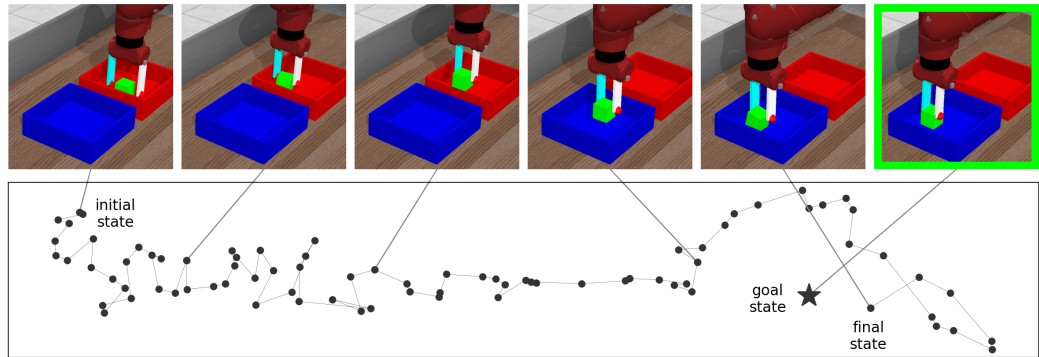

Figure 14: **Visualizing the learned representations.** *(Top)* We show five observations from the bin picking task, as well as the goal image. *(Bottom)* A TSNE embedding of the image representations $\phi(s, a)$ learned by Contrastive RL (NCE). Note that different parts of the task (e.g., reaching, picking, placing) are well separated in the learned representation space.

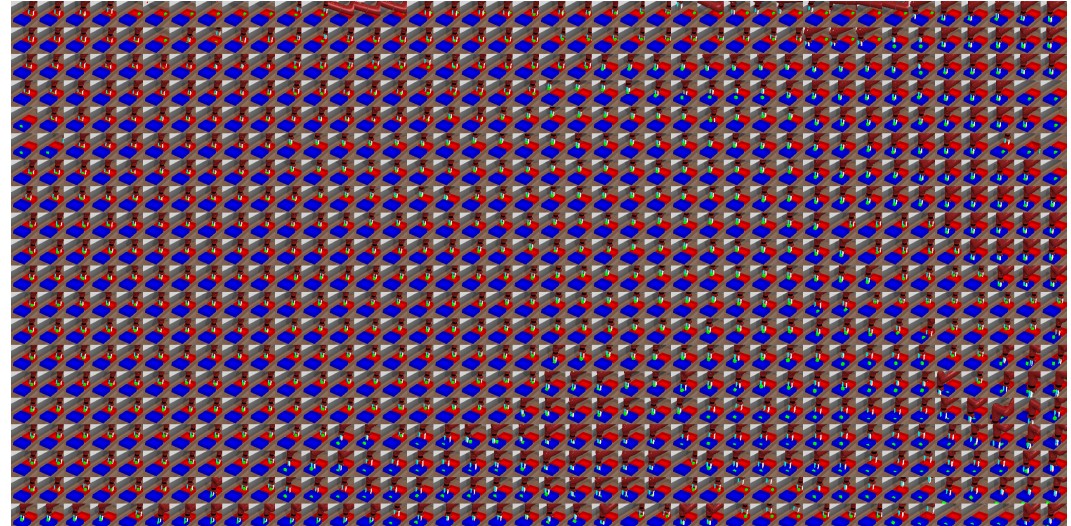

Figure 15: Visualizing the image representations learned by our method on the `sawyer bin`. We compute a TSNE embedding of the representations, and then align the embeddings to a grid using RasterFairy [65].

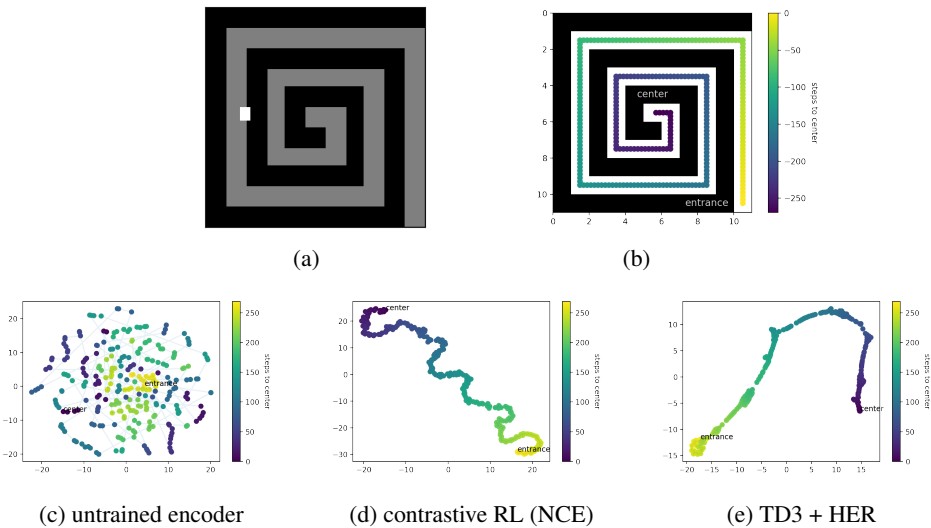

(a)                                                    (b)

(c) untrained encoder          (d) contrastive RL (NCE)          (e) TD3 + HER

Figure 16: **TSNE embedding of representations** $\phi(s, a)$. *(a)* Using the `point Spiral11x11` task, *(b)* we generated image observations at 270 locations throughout the maze. We computed the state-action representations of these images, using action = (0, 0). *(c, d, e)* A TSNE embedding of these representations reveals that the untrained encoder does not capture the structure of the environment, whereas both our method and the TD3 + HER baseline do capture the maze structure.

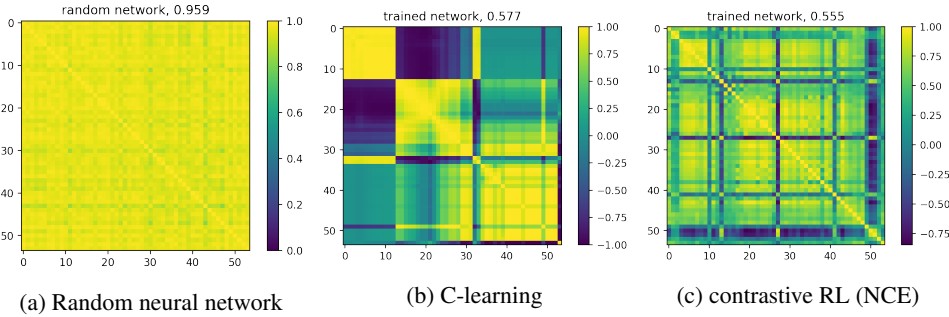

(a) Random neural network          (b) C-learning          (c) contrastive RL (NCE)

Figure 17: **Analyzing the gradients.** We plot the cosine similarity between the (normalized) gradients of the critic function with respect to the goal images. An untrained network has high gradient similarity, meaning that updates to one state/task affect the networks predictions at many other states/tasks, a phenomenon that prior work has identified as being detrimental to learning [2, 70, 135, 138]. Our method converges to a network where gradients at one state have a low similarity with gradients at other states. A similar plot showing gradients with various state inputs shows a similar effect.

# G  Failed Experiments

1. *Representation normalization*: We experimented with many ways of normalizing the learned representations, including L2 normalization, scaling the representations by a learned temperature parameter, and applying an elementwise tanh activation. None of these modifications consistently improved performance.

2. *Momentum encoder*: Prior contrastive learning methods have found it useful to use a target encoder or momentum buffer. We experimented with many similar tricks, including using a momentum buffer for goal representations, sampling some fraction of goal representations from a fixed random distribution, increasing the learning rate for the state encoder's final layer, and decreasing the learning rate for all layers in the goal encoder. None of these tricks consistently improved performance.

3. *Frame stacking* – This tended to decrease performance slightly.

4. *Loss scaling* – Our contrastive RL (NCE) method uses the negative label much more often than the positive label. We tried scaling the loss terms so that the negative and positive examples received the same total weight, but found this had no effect on performance.