# OpenReview forum: "Contrastive Learning as Goal-Conditioned Reinforcement Learning"
_NeurIPS.cc/2022/Conference — NeurIPS 2022 Accept_

### Official Review · Reviewer_87oA · 2022-07-04

**Rating:** 7
**Confidence:** 4
**Soundness:** 3 good
**Presentation:** 4 excellent
**Contribution:** 3 good

**Summary:**

The paper presents a novel view of how contrastive learning can be used by itself, and without any additional RL training on top, to learn goal conditioned policies from pre-collected data. This is achieved by using the similarity learned during contrastive learning as a Q-function, that is then used to improve a policy in the policy gradient step. It is also shown, via a proof, that the contrastive objective estimates the Q-function, and some additional convergence guarantees are provided. The authors well motivate their approach, compare it extensively with related work on learning goal-conditioned policies, as well as works that use an additional representation learning objective on top of a RL algorithm. The paper shows a good comparison between prior work and the proposed method on a range of goal-conditioned RL tasks that have been used in these prior works. The overall conclusion is that a much simpler method, like the one presented, can perform similarly well, and in some cases even better, than more complicated baselines.

**Questions:**

One question I would be interested in, is how well the method would perform in partially observable, large environments where the agent has a first-person view.Goal conditioning from pixels seems to be much easier in situations with a fixed camera covering the majority of the possible states, but it seems that the difficulty of the problem quickly grows when we move to first-person views in large environments with partial observability.
I’m also wondering how well the trained method generalizes zero-shot to changes in the color of the background, the setup of the hand (i.e. having the base of the sawyer arm on the left side of the image) or to new types/colors of objects.

In Figure11 in the appendix, the colour seems to be wrong way round in the plot of the maze, as the positions in the centre have the colour corresponding to the highest "steps to centre". The colour for the tSNE plots seems to be correct though.

Two typos:
Line 154: “The negative” -> For the negative
Line 215: “interation” -> iteration

**Limitations:**

 The limitations have not been added to the author’s submission checklist yet, but are addressed in the actual paper.

**Strengths And Weaknesses:**

### Strengths
1. The paper presentation and clarity is excellent and of high quality. The authors have performed an extensive evaluation of their method and compared it against two sets of baselines, one based on representation learning methods and one based on learning goal-conditioned policies. The authors describe their method, motivation and results very clearly and do a very good job at relating it to prior work.
2. Their method is grounded in theory, though I cannot claim I could follow 100% of their proofs and arguments as I am not an expert on the theoretical side.
3. Simplicity of the method, while simultaneously performing similarly well to other methods, and in some cases outperforming strong and more complicated baselines.
4. The authors present an original perspective on how contrastive learning can be used to learn goal-conditioned policies directly.

I did not find any noteworthy weakness to the paper other than some typos and some more questions about the limits of this method that I have included in the next section with Questions.

---

> ### Author Response · Authors · 2022-08-02
> **Response to 87oA**
>
> We thank the reviewer for their detailed and thorough review and for the suggestions for improvement. To improve the paper, we have run additional experiments to answer the reviewer questions, studying how contrastive RL copes with partial observability and whether the learned policy generalizes zero-shot.
>
> > Can the algorithm handle first-person views and/or partial observability
>
> We ran an additional experiment to study how contrastive RL copes with moving cameras and partial observability, finding that it can solve such tasks, though performance is worse than with a fixed camera. To study this question, we modified the sawyer push task so that the camera tracks the hand at a fixed distance, as if it were rigidly mounted to the arm. At the start of the episode, the scene is occluded by the wall at the edge of the table, so the agent cannot see the location of the puck. Nonetheless, contrastive RL (NCE) successfully handles this partial observability, achieving a success rate of around 35%. Please see the new videos at the bottom of the project website (https://contrastive-rl.github.io/) and new Appendix F.6 and Figure 11 for details.
>
> > generalizes zero-shot to changes in the color of the background, the setup of the hand, or to new types/colors of objects.
>
> We ran an additional experiment to study the robustness of the policy learned by contrastive RL (NCE), finding that it is robust to object color and setup of the hand but is not robust to the color of the table background. The new experiment, which we describe in new Appendix F.7, shows that changing the object color from black to red has little effect, as does moving the initial hand position towards the camera. However, changing the table color from white to yellow decreases the success rate from 78% to 0%.
>
> > In Figure11 in the appendix, the color seems to be wrong way round
>
> Thanks for catching this! We have fixed this.
>
> > typos
>
> Thanks for catching these! We have fixed them.
>
> We hope that these responses have addressed the reviewer's questions. **Does the reviewer have additional questions or concerns?**

---

> > ### Comment · Reviewer_87oA · 2022-08-08
> > **Response to authors**
> >
> > I would like to thank the authors for their detailed response and for running the additional experiments. I find the additional experiment and video with the hand-tracking camera interesting, thank you for running that.

---

### Official Review · Reviewer_hUWA · 2022-07-11

**Rating:** 8
**Confidence:** 3
**Soundness:** 3 good
**Presentation:** 4 excellent
**Contribution:** 4 excellent

**Summary:**

This paper proposes leveraging contrastive learning to action-labeled trajectories, where the learned representation will connect to the goal-conditioned value functions. Specifically, they utilize contrastive learning to estimate the Q-function. The critic function (parameterized by the inner-product between representation) in contrastive learning can also be applied as the critic function in actor-critic algorithms in the RL context. Theoretical analysis of the convergence guarantees has been provided in the meantime. The empirical results suggest that the proposed framework can lead to better performances on goal-conditioned benchmarks compared with prior methods, including methods using goal-conditioned RL and different representation learning strategies.

**Questions:**

Please note that below are only for questions and potential discussions. There is no need to rerun experiments during the rebuttal phase.

**1. About the required goal**

In this work, the goal is the state/observation given by the task. Can this requirement be looser?

**2. About the generalization abilities**

Can the learned representation be generalized to a set of diverse tasks with only a smaller number of interaction data?



**Limitations:**

The authors mentioned the limitations in the paper, mainly about how to generalize the work into other RL settings. I think this would be the potential direction for future works. I raise a few questions, which are more like vague points instead of limitations. I think this is a decent paper and I would vote for acceptance.

**Strengths And Weaknesses:**

### *Strengths*

#### **1. Originality and significance**

Although this work is built upon existing literature on goal-conditioned RL and contrastive learning (in RL), it has enough novelty since it provides a clear direction to link the reward maximization and learning representation using contrastive learning. The way offered in the paper to directly use contrastive learning to perform goal-conditoned RL is very simple and efficient. This work may have the potential to benefit the RL, representation learning, and robotics communities.

#### **2. Relevance**

The authors discuss most related works, including goal-conditoned RL, representation learning for RL, and contrastive learning (for RL). Detailed comparison has also been given in the associated sections.

#### **3. Algorithms, theories, and evualation**

The designed algorithm is decent, and I feel this can broadly apply to other RL regimes. The convergence guarantee makes it more solid. The experiments part is also very solid, including most relevant baselines and benchmarks. The appendix H on failed experiments is also very helpful for the community.

### *Weaknesses*

Please note that below are only for questions and potential discussions. There is no need to rerun experiments during the rebuttal phase.

#### **1. About experiments**

It seems that, in some cases, improvements compared with C-learning are minor (Fig. 5). Can the authors give some explanation. Meanwhile, why not compare with other contrastive learning approaches in RL mentioned in the early sections?

#### **2. About other common challenges in goal-conditioned RL**

Can this approach tackle common challenges in goal-conditioned RL, especially for off-line datasets? e.g., the generalization ability to different distributions.

#### **3. Minor typos**

-> Line 107: $\pi\left(\tau \mid S_{t}\right)$ -> $\pi\left(\tau \mid S_{g}\right)$;

-> Eq. 4: consider using another color, the current $v^{+}$ is hard to recognize when print out the paper to read.

---

> ### Author Response · Authors · 2022-08-02
> **Response to hUWA**
>
> We thank the reviewer for their very thorough and detailed review and welcome the suggestions for improvement.  To improve the paper, we have run two additional experiments, studying how contrastive RL can be applied to the offline RL setting and whether the learned representations can be transferred to solve new tasks.
>
>
> >  why not compare with other contrastive learning approaches in RL mentioned in the early sections?
>
> We have already compared our algorithm to CURL (Fig 4, "x" markers), a method that performs contrastive learning to extract representations and then uses those representations for RL tasks. While CURL was designed for maximizing a single task reward, we adapted it to the goal-conditioned RL setting by combining it with the strongest baseline from our experiments (TD3+HER).
>
> > Can this approach tackle common challenges in goal-conditioned RL, especially for offline datasets?
>
> As suggested by the reviewer, we applied contrastive RL to a benchmark suite of offline datasets. The results, described in new Appendix F.2 and Appendix Table 3, show that contrastive RL is competitive with (and sometimes better than) IQL, a recent paper that claims state-of-the-art results. Compared with recent prior methods (RvS-G, Decision Transformer) that do not use TD learning (IQL uses TD, but contrastive RL does not), contrastive RL improves the results by a wide margin.
>
> > Line 107: π(τ∣St) -> π(τ∣Sg);
>
> Fixed.
>
> > Eq. 4: consider using another color; the current v+ is hard to recognize when printing out the paper to read.
>
> We have switched this to a darker shade of green.
>
> > In this work, the goal is the state/observation given by the task. Can this requirement be looser?
>
> Yes, the goal could be any metadata or label provided to the observation. For example, image observations could be labeled with natural language. Then, by applying the same contrastive RL algorithm as in this paper, we could learn a policy where the goal is natural language ("the bedroom is clean"). We haven't tried this yet, but the success of contrastive learning on prior multimodal applications  (e.g., CLIP [1], Dalle-2 [2]) bodes well for this potential application.
>
> > Can the learned representation be generalized to a set of diverse tasks with only a smaller number of interaction data?
>
> We ran an additional experiment to study this question, finding that the representations can sometimes be transferred to accelerate solving a new task. We applied contrastive RL (NCE) to three image-based tasks for 1M environment steps, and then used those representations for solving new tasks. The results, shown in the new Appendix Figure 10, indicate that representations from "sawyer bin" can help in solving the "sawyer push" task and that representations from "fetch push" can help in solving the "sawyer bin" task; other combinations work no better than randomly initializing the representations.
>
> We hope that these responses have addressed the reviewer's questions. **Does the reviewer have additional questions or concerns?**
>
> [1] Radford, Alec, et al. "Learning transferable visual models from natural language supervision." International Conference on Machine Learning. PMLR, 2021.
>
> [2] Ramesh, Aditya, et al. "Hierarchical text-conditional image generation with clip latents." arXiv preprint arXiv:2204.06125 (2022).

---

> > ### Comment · Reviewer_hUWA · 2022-08-05
> > **Response**
> >
> > Thanks for the detailed and thoughtful response. My concerns have been addressed by the rebuttal and revision. I think it is a strong submission.

---

### Official Review · Reviewer_Txuo · 2022-07-11

**Rating:** 7
**Confidence:** 3
**Soundness:** 3 good
**Presentation:** 3 good
**Contribution:** 3 good

**Summary:**

The authors have proposed a well-motivated contrastive learning method for solving goal-conditioned RL tasks, thus removing the need for any auxiliary losses or data augmentations for representation learning. The authors then show how this contrastive objective leads to a critic implicitly learning the Q-value function, and empirically demonstrate the effectiveness of their simple approach on many tasks.

**Questions:**

1. typos:
    - line 141: “The ~~objective~~ expected reward objective”
    - line 188: “critic is parametrized as an *inner* product” - but in Algorithm 1, the comment says that you performed an outer product.
    - line 268: “One” → “On”
    - line 879: “~~has~~ uses”
2. For the fetch and sawyer tasks, what is the dimension of goals (state-based)? I am assuming that for image-based, it is 64x64x3, right?
3. Could the authors change the color palette for the plots in Figure 5? Right now, it’s extremely difficult to judge which line corresponds to which baseline, especially on print.

**Limitations:**

Yes, the authors have addressed the major limitations of their method.

**Strengths And Weaknesses:**

# Strengths

1. The proposed method simplifies the representation learning + planning in RL problem, by using the inner product of learnt representations as a correlate for the Q-value function.
2. This is especially useful for the pixel-based environments, for which the authors have shown performance gains in many continuous-control tasks.
3. I also appreciate that the authors mentioned certain failed experiments in Appendix H, which help in better understanding of the proposed method.
4. The paper presents theoretical motivation for their method and show how this contrastive objective relates to a critic in deep RL methods.
5. Thorough comparison with several related works.

# Weaknesses

1. It is unclear why *random goals* are sampled to train the actor loss? Can the authors shed more intuition on this decision?
2. Can the authors also comment if there’s some minimum goal dimension, after which contrastive RL starts performing better than the model-based baseline?
3. In Figure 5, while contrastive RL and its variants surely perform better in simpler state-based tasks, however, looking at the plots for image-based observation tasks (Fig 5(b)), it seems that C-learning alone does quite well, especially in the challenging tasks of Sawyer bin. Also given that in fetch push (Fig 5(b)), contrastive RL (NCE + C-learning) gets a huge boost over other the baseline, I think that’s also because of the addition of C-learning pipeline only.

---

> ### Author Response · Authors · 2022-08-02
> **Response to Txuo**
>
> We thank the reviewer for their thorough and detailed review and welcome suggestions for improvement. We have revised the paper to incorporate the reviewer's feedback. We have also added a new offline RL experiment: contrastive RL is competitive or better than prior offline RL methods on an offline goal-conditioned benchmark suite. Below, we will answer the reviewer's questions.
>
> > It is unclear why random goals are sampled to train the actor loss. Can the authors shed more intuition on this decision?
>
> While in theory the distribution of goals for the actor loss shouldn't affect the optimal policy, in practice we found that using random goals in the actor loss works much better than using future goals. We have revised the paper to include an ablation experiment showing this (Appendix Figure 9). Our intuition is that the future goals are too easy to reach and that random goals are required to train the policy to reach more challenging goals. One important exception is for the new offline RL experiments (see Appendix F.2), where we found that sampling future goals for the actor loss improved the performance.
>
> > Can the authors also comment if there’s some minimum goal dimension, after which contrastive RL starts performing better than the model-based baseline?
>
> We ran an additional experiment to answer this question, varying the goal dimension on the state-based "sawyer push" task. These results, now included in Appendix Figure 8, show that contrastive RL starts to perform better than the model-based baseline when the goal dimension is 4 or larger.
>
> > In fetch push (Fig 5(b)), contrastive RL (NCE + C-learning) gets a huge boost over the baseline because of the addition of C-learning pipeline.
>
> We agree with this interpretation, but note that it can also be interpreted as saying that taking the prior method (C-learning) and combining it with ours (contrastive RL) makes the prior method much better.
>
> Regardless of whether this is viewed as X+Y or Y+X, it remains an open question why this combination would work much better, given that both C-learning and contrastive RL are performing some sort of contrastive learning (see Sec. 4.6). One hypothesis is that contrastive RL helps very early on during training, when the TD updates of C-learning end up "backing up" bad predictions; then, once the classifier has stabilized, the TD updates improve the sample efficiency.
>
> > Typos
>
> We've fixed these.
>
> > For the fetch and sawyer tasks, what is the dimension of goals (state-based)?
>
> The state and goal dimensions for these tasks are:
> * fetch_reach: dim(state) = dim(goal) = 10
> * fetch_push: dim(state) = dim(goal) = 25
> * sawyer_push: dim(state) = dim(goal) = 7
> * sawyer_bin: dim(state) = dim(goal) = 7
>
> > I am assuming that for image-based, it is 64x64x3, right?
>
> Yes, both the observation and the goal images are 64x64x3.
>
> > Could the authors change the color palette for the plots in Figure 5?
>
> We have updated the figure to use different colors (not different shades of blue).
>
> We hope that these responses have addressed the reviewer's questions. **Does the reviewer have additional questions or concerns?**

---

> > ### Comment · Reviewer_Txuo · 2022-08-06
> > **Response**
> >
> > I appreciate the authors taking the time to respond to my comments and making appropriate changes. I agree with the detailed responses and recommend an easy "Accept" for this paper.

---

### Meta-Review · Area_Chair_CvhA · 2022-08-25

**Recommendation:** Accept
**Confidence:** Certain

**Metareview:**

How to design RL algorithms that directly acquire good representations? This paper gives an answer that contrastive representation learning can be cast as a goal-conditioned RL using the inner product of learned representations.
The technical novelty of this paper is sound, with the thorough theoretic motivation of the proposed method and solid experiments. The presentation of this paper is also satisfactory.
All the reviewers provided positive feedback on this paper. I also enjoy reading this paper.


**Award:**

No

---

### Decision · Program_Chairs · 2022-09-14

Accept